# Hybrid Nanoplatforms Based on Photosensitizers and Metal/Covalent Organic Frameworks for Improved Cancer Synergistic Treatment Nano-Delivery Systems

**DOI:** 10.3390/molecules30040884

**Published:** 2025-02-14

**Authors:** Aviwe Magadla

**Affiliations:** Department of Chemical and Physical Sciences, Faculty of Natural Sciences, Walter Sisulu University, Nelson Mandela Drive, Mthatha 5117, South Africa; amagadla@wsu.ac.za

**Keywords:** photosensitizer (PS), covalent organic frameworks (COFs), metal–organic frameworks (MOFs), nanomaterials, drug delivery systems, photodynamic therapy (PDT), photothermal therapy (PTT)

## Abstract

Researchers have extensively investigated photosensitizer (PS) derivatives for various applications due to their superior photophysical and electrochemical properties. However, inherent problems, such as instability and self-quenching under physiological conditions, limit their biological applications. Metal-organic frameworks (MOFs) and covalent organic frameworks (COFs) represent two relatively new material types. These materials have high surface areas and permanent porosity, and they show a tremendous deal of potential for applications like these. This review summarizes key synthesis processes and highlights recent advancements in integrating PS-based COF and MOF nanocarriers for biomedical applications while addressing potential obstacles and prospects.

## 1. Introduction

The domain of drug delivery systems has garnered considerable interest in recent years, propelled by the demand for more efficient and tailored therapeutic approaches. These delivery devices function as nanocarriers, swiftly carrying medication to the targeted area while preventing its quick elimination or degradation. Drug delivery systems face challenges, including inadequate solubility, diminished bioactivity, and suboptimal targeting. Numerous inorganic (e.g., iron oxide nanoparticles (NPs), noble metal NPs, quantum dots) and organic (e.g., liposomes, polymers, dendrimers) nanomaterials have been engineered as nanocarriers, each possessing distinct advantages and disadvantages [1,2,3]. Among numerous advanced materials, metal and covalent organic frameworks (MOFs and COFs) and organic small molecular photosensitizers (PS), such as indocyanine green (ICG), porphyrins, phthalocyanines (Pcs), BODIPY, and other dyes, including hybrid composites, have emerged as promising candidates for improving drug delivery mechanisms [4,5,6,7,8,9,10,11,12].

MOFs and COFs are customizable and adjustable functional crystalline porous materials investigated for applications such as catalysis, chemical sensing, water harvesting, gas storage, and separation [13,14,15,16]. COFs create two- or three-dimensional structures by interacting with organic precursors, forming covalent bonds, and yielding porous organic materials. MOFs, COFs, and PS possess unique characteristics for drug delivery. MOFs, also referred to as porous coordination polymers (PCPs), are crystalline coordination polymers. Due to the highly organized porosity and tunable properties of MOFs, they serve as an exceptional substrate for encapsulating various medicinal agents [17,18,19,20]. Their inherent characteristics, such as extensive surface area, enhanced stability, and ability to interact with diverse organic linkers, facilitate controlled drug release and improved bioavailability. They offer several benefits compared to traditional nanocarriers: (i) they can be engineered to create specific structures with varying shapes, sizes, and chemical characteristics, facilitating the incorporation of diverse therapeutic agents with distinct functionalities; (ii) their extensive surface area; significant porosity, consistent pore size, and volume contribute to elevated loading efficiency and selective transport; (iii) they are biodegradable owing to their unstable metal and ligand coordination bonds; and (iv) surface functionalization can improve colloidal stability and prolong the blood circulation time [21,22,23,24].

Numerous hybrid MOFs and COFs exhibit exceptional features, and the deficiencies of individual MOFs and COFs can be mitigated by creating hybrid MOF and COF delivery systems. In recent years, investigations have been conducted on the transport of various biomolecules using COF and MOF nanocarriers, as mentioned earlier. Anticancer agents, such as doxorubicin [25,26], topotecan [27], and 5-fluorouracil [28,29], to mention a few, have been delivered intracellularly via MOFs. PS-functionalized MOFs and COFs have been applied as anticancer agents for photodynamic treatment (PDT) [30,31,32].

PS are agents that absorb light at a specific wavelength and transform it into usable energy. Due to their exceptional photophysical and electrochemical capabilities, PS derivatives are acknowledged for their photophysical characteristics and versatility in various applications, including catalysis, biosensing, gas storage, solar cells, and biomedical uses [33,34,35]. However, their biological applicability, particularly in cancer therapy and detection, is significantly constrained by inherent limitations, including self-quenching, weak absorption in the biological spectrum window, and inadequate chemical and optical stability.

In the past two decades, numerous natural and synthetic dyes, such as PS, have been examined in vitro and in vivo in PDT research. When combined with MOFs and COFs, PS can enhance the therapeutic efficacy of drug delivery systems through methods such as PDT, which utilizes light to initiate drug release or augment absorption by targeted cells. Efforts have been undertaken to associate existing molecular PS with MOFs to address the aggregation problem for biological applications [30,36]. Incorporating these two elements into composites facilitates the development of multifunctional platforms capable of addressing the constraints of conventional drug delivery, including low solubility, rapid metabolism, and nonspecific targeting [37,38,39]. Administering these biologically significant molecules as biomolecular therapeutics presents a novel approach to disease treatment.

This succinct review article provides an overview of the various synthetic methodologies used to improve the design of MOFs, COFs, and PS hybrid composites for drug delivery purposes. We will thoroughly examine the PS-based MOF/COF targeting techniques used in tumor-targeted therapy in recent years, along with an analysis of potential synergistic effects in medicinal applications. This work seeks to serve as a significant reference and source of concepts for targeted therapy utilizing PS-based MOF or COF materials, while also encouraging further investigation of their potential for cancer treatment.

## 2. Synthesis and Methods

Metal ions or clusters self-assemble with organic ligands to produce repeating building blocks, the process by which photo-responsive MOFs are synthesized. A variety of factors influence their properties during the synthesis process. These elements generally encompass the photoelectronic properties of the metal ions and organic ligands, the solvent employed, the crystal formation speed, the process’s length and temperature, and the crystal formation [40,41]. Researchers have devised various conventional synthesis methods for MOFs, including hydro/solvothermal, sonochemical, microwave, mechanochemical, and electrochemical methods. The hydro/solvothermal method is highly favored for synthesizing photo-responsive MOFs due to its cost-effectiveness, simplicity, and convenience [40,42]. In the same vein, there are numerous methods for synthesizing COFs. Conventional COF techniques include microwave [43], solvothermal [44], ionothermal [45], mechanochemical [46], and vapor-phase-assisted [47]. Assessing these factors and determining the most appropriate approach for the desired results is crucial. The precise selection of a method should be determined by the characteristics of the monomers and the necessary bonding conditions. Ultimately, the ideal MOF/COF synthesis method is determined by the specific requirements of the synthesis, including the desirable structure, scalability, reaction time, impurity removal, pressure, temperature, solvent selection, and potential for industrial production.

### 2.1. MOF Drug Incorporation Delivery Pathways

Drug loading tactics for MOFs have the distinguishing characteristics of having a large surface area and structural divergence, which make it easier to load drugs on the exterior or inside the pores using a variety of loading strategies. The one-step strategy and the two-step method are two forms of drug-loading strategies that are utilized frequently [48].

#### 2.1.1. One-Step Synthesis

This technique directly incorporates medicinal agents with the MOF during their synthesis. This technique employs uniform distribution and substantial drug-loading capability. Nonetheless, regulating particle size, shape, and physicochemical properties of existing MOFs is difficult. Furthermore, it is imperative to implement additional measures to guarantee that medicine remains undamaged during the production process [49].

The one-pot approach involves the co-precipitation of the medication with the MOF during synthesis. As a result, the pores of MOFs evenly distribute drug molecules [49]. The one-step synthesis approach accomplishes the entire synthesis process under a singular reaction state, eliminates the need for separation and purification of intermediate products in multi-step synthesis, streamlines the synthesis pathway, and enhances synthesis efficiency. Zheng et al. [50] proposed an innovative method integrating MOF production with molecular encapsulation in a single-step procedure. They demonstrated that zeolitic imidazolate framework (ZIF) crystals can contain substantial medication and dye molecules. The crystals uniformly disperse the molecules and allow for the adjustment of their concentrations. They showed that ZIF-8 crystals filled with the cancer-fighting drug DOX work well as drug delivery systems in cancer treatment because they release the drug when the pH level changes. Their efficacy on breast cancer cell lines surpasses that of free DOX. Their work demonstrates that the one-pot technique creates new opportunities to develop multifunctional delivery systems for many applications.

##### Drugs as Organic Linkers for MOFs

Drugs or their prodrugs may operate as organic ligands to form MOFs and act as MOF reservoirs by coordinating their specific functions with designated metal ions [51]. Xu, Zhen, and colleagues reported the invention of a new nMOF, Hf-TP-SN, with an X-ray-triggerable 7-ethyl-10-hydroxycamptothecin (SN38) prodrug for synergistic radiation and chemotherapy. One of the most extensively studied uses of MOFs in PDT is the incorporation of organic PS molecules as linkers into the MOFs’ structure [52]. Selective MOFs have inherent ROS production capabilities even without organic dye PS molecules that are embedded as linkers on the nanoconjugate surface or enclosed in the channels [53]. Unlike MOFs containing PS molecules as linkers, which generate ^1^O_2_, these MOFs generate ^1^O_2_ and hydroxyl radicals (∙OH-) by catalyzing Fenton-type conversions of H_2_O_2_ to yield the hydroxyl and superoxide radicals and form ^3^O_2_ from H_2_O_2_. In situ, the production of ^3^O_2_ from H_2_O_2_ is commonly used to treat TME hypoxia [54], Figure 1.

A study by Truong Hoang Q and colleagues also introduced nanoscale zirconium-based p-MOFs (PCN222) as safe and effective nanosonosensitizers. Polyethylene glycol (PEG)-coated PCN-222 (PEGPCN) was infused with the pro-oxidant agent piperlongumine (PL) to facilitate tumor-targeted chemo-photodynamic combination therapy. PEG-PCN and PL-incorporated PEG-PCN (PL-PEG-PCN) exhibited significant colloidal stability in biological media. Moreover, nanoscale PL-PEG-PCN was effectively internalized by breast cancer cells, resulting in a substantial enhancement of ROS formation with ultrasonic exposure [55], (Figure 2).

##### Co-Crystallization

Laboratory research and industry use co-crystallization to load medicines into MOFs. Minimal reaction conditions can create a 3D supramolecular structure containing the medication. Co-crystallization improves drug solubility and loading without affecting their physicochemical properties. Most importantly, co-crystallization does not alter the drug’s physicochemical properties, which can improve solubility and loading. Using slurry and solvent evaporation, Hao Cheng and coworkers [56] co-crystallized mannitol with CaCl_2_. They compared tablet performance between co-crystal and mannitol. The bonding area-bonding strength (BA-BS) model associated tabletability differences with crystal structures. The produced co-crystal has a 1:1:2 molar ratio of mannitol, CaCl_2_, and water (i.e., mannitol·CaCl_2_·2H_2_O). Mannitol molecules link the Ca^2+^ in the co-crystal through an endless coordination network, forming a classic MOF structure. Compared to β-mannitol, MOF-based co-crystals showed better tabletability (2 times higher tensile strength) and less tendency to laminate (3 times higher minimum compaction pressure). Due to stronger intermolecular contacts, the co-crystal’s greater BS improved tabletability. The lowered lamination propensity was due to lesser in-die elastic recovery than β-mannitol.

#### 2.1.2. Two-Step Synthesis

The two-step synthesis method optimizes the reaction conditions at each step to obtain the best reaction results. This phased synthesis strategy is conducive to selecting the appropriate solvent, temperature, and reaction time to improve the synthesis efficiency and product quality. When drugs’ molecular dimensions are smaller than MOFs’ pore diameter, this technique often predicts their containment within the scaffolds through hydrogen bonding, contacts, or other host–guest interactions. It is anticipated that larger drug molecules with opposing charges will be adsorbed by MOFs via electrostatic interactions [57,58].

##### Impregnation

Due to their porous nature and accessibility to metal ions and tiny molecules, MOFs can be impregnated with precursors by diffusion/deposition. The MOF solids are immersed in a precursor-laden solution for impregnation. The adsorbed precursors become the final active species or undergo further reactions (such as reduction, decomposition, or other chemical processes) to form new functional phases in the MOF matrices. Zuly’s and associates [59] used solvothermal synthesis to create AgNPs@MOF-808 nanocomposites, which are used for drug delivery. They utilized a zirconium-based MOF with benzene-1,3,5-tricarboxylic acid linkers, also known as MOF-808. MOF-808 (72 h) had the highest crystallinity with a nano-MOF particle size of around 77–277 nm. Ag⁺ was incorporated into AgNPs at different AgNO_3_ concentrations (0.01, 0.05, 0.2, and 0.4 mmol), utilizing MOF-808 as a template and DMF as a mild reductant. AgNPs@MOF808 (0.4) dispersed a maximum of 5 nm nano spherical AgNPs within the pores, accompanied by an octahedral MOF of approximately 130 nm.

##### Covalent Binding

Within the MOF structure framework, covalent bonds are created by utilizing organic linkers and inorganic metal clusters during covalent binding. Even though the method can insert a wide variety of cargoes into MOFs, there are frequently issues with the pharmaceuticals slowly leaking due to the relatively weak contact force between the drugs and MOFs [60]. The potential for the medications to cause leakage creates a problem. In addition, some MOF derivatives can break down and clump together in vivo, while loaded medications are known to experience the burst effect in these settings. X. Liu and associates [61] developed a simple yet effective bilayer coating method to mitigate these issues as a conventional, dual-phase process component. The bilayer-coated MOF NU-901 also dispersed well in biologically relevant fluids like buffers and cell growth media throughout the study. Adding the coating makes the drug-loaded MOFs more stable in water by stopping drug leakage and MOF particles from sticking together.

### 2.2. PS-COF-Based Synthesis

Porphyrins are abundant in nature and play essential roles in many living creatures, including cytochromes, hemoglobin, and chlorophyll. Metalloporphyrin complexes, known as “life pigments”, compose these biological entities and play critical roles in life processes [62,63]. The unique properties of their tetrapyrrole macrocycles determine the porphyrins’ functional prowess, including biochemical, enzymatic, and photochemical capabilities. As such, most research into synthesizing PS-based COFs (PS-COFs) has focused on integrating porphyrins with COFs (p-COFs) for use in biological applications. The preparation of p-COFs is similar to those of other COFs. The construction of p-COFs frequently employs imine, triazine, and borate condensation reactions [64]. The thermodynamic regulation allows for the reversibility of the ongoing reactions [65,66]. Processes can self-correct and self-heal during the production of reversible covalent bonds [8,67,68]. These processes produce ordered and thermodynamically stable p-COFs [69,70].

### 2.3. Significance of PS-COF and MOF-Based Nanohybrids in Cancer Treatment

A significant disadvantage of relying solely on PS-based PDT is that additional PS can readily leave the vasculature and enter the tumor, which is absorbed by cancer cells and, upon contact with light, causes cytotoxicity. Furthermore, extensive PS usage frequently leads to inadequate drug localization, which fuels medication toxicity. Additionally, PS varies in solubility, and some PS need nanocarriers or other compounds for cancer treatment [71]. At the moment of illumination, the PS’s placement significantly impacts the effects of PDT. The combination of several functional moieties, which is frequently advantageous in drug delivery inside PS-MOF and PS-COF nanohybrids, is made possible by the varied structural design achieved by assembling predesigned building units. When excited by light (Figure 3), the capacity to be built with PS metal ions or organic complexes that can generate ROS or cytotoxic ^1^O_2_ to kill cancer cells is another benefit of PS-MOFs and PS-COFs [72,73].

In particular, by avoiding aggregation-induced quenching and overcoming the poor water solubility of the high light-conjugated PS, PS-COFs, and PS-MOFs with staggered stacking of PS, we improve photoluminescence performance and ROS generation. The accumulation of the nanodrug system at tumor sites is enhanced by the drug-loading system’s enhancement of the permeability and retention (EPR) effect at tumor sites through nanocrystallization. To enable circulation, aggregation, and retention via the EPR effect in the tumor microenvironment, the size of the nanoscale of PS-MOFs and PS-COFs is a crucial property during passive targeting [74].

Furthermore, the porosity characteristics and π–π stacking interactions of PS-MOFs and PS-COFs facilitate the effective loading of diverse organic small molecules, hence augmenting the synergistic therapeutic benefits of PDT and PTT. However, the non-simultaneous occurrence of PDT and PTT can hinder fully realizing the synergistic processes. By increasing the local temperature of the cancer tissue, photothermal therapy can improve oxygen transport and blood circulation, which helps to reduce the hypoxia that prevents PDT. Therefore, creating a single-light-source-activated photosensitive material that can give PDT and PTT is desirable and worthwhile [75].

## 3. Biocompatibility

### 3.1. PS

The optimal biocompatible PS should have high light absorption coefficients, especially for long-wavelength near-infrared radiation, to help them penetrate deeper into tissues; low photobleaching quantum yields; high intersystem crossing efficiencies; low toxicity when light is not present; and, finally, the right balance of hydrophilic and hydrophobic properties to help them stick to tumors [76,77,78]. The predominant alterations to the PS-based complexes encompass incorporating water-soluble moieties, enhancing optical characteristics, attaching targeting entities to facilitate accumulation in neoplastic cells, and extending therapeutic efficacy. In recent decades, researchers have thoroughly examined various methods for synthesizing PS derivatives and altering their photophysical and photobiological characteristics [31]. In the field of PDT, there have been documented clinical trials that have utilized PS [79,80,81,82,83,84,85].

The conjugation or integration of PS into nanostructures facilitated their application in nanomedicine. The third generation of PS has a lot of benefits, such as the ability to accomplish more than one thing, like targeting tumor tissues specifically and improving the efficiency and selectivity of intracellular delivery with PS [4,31]. In the fourth generation of PS, porous carriers are used, and these include COFs and MOFs.

### 3.2. PS-Based MOF Composites

Metallic compounds employed in biomedical applications are frequently examined for their possible long-term consequences. The elevated reactivity and diminutive size of MOFs raise safety issues concerning human health. Researchers have conducted limited toxicity studies to evaluate the cytocompatibility of MOFs and their hybrid analogues, which has led to an inadequate characterization of their toxicity. MOFs may accumulate within cells due to their reduced size. Metal accumulation over time may pose safety risks, particularly if MOFs are employed as long-term medicinal delivery methods [86,87]. Additionally, further study is required to create and construct innovative MOF nanocomposites with improved stability, biocompatibility, and therapeutic performance because MOFs’ pharmacokinetics, degradation mechanisms, and toxicity are poorly understood [88,89]. Ultimately, research efforts in the field of MOF-based composite treatments should result in the development of more sophisticated and programmable MOF nanocomposites that support multiple functionalization for the combination of diagnosis and various therapies in a single MOF, aiming for the effective eradication of diverse primary and metastatic tumors, as well as the prevention of recurrence, with minimal toxicity and side effects. The advancement of the discipline also depends on the validation and standardization of testing protocols [90]. Therefore, conducting comprehensive in vitro and in vivo toxicity assessments is essential to guarantee biocompatibility. A correlation between in vitro and in vivo research is necessary to determine the appropriate dosage within safety parameters.

### 3.3. PS-Based COF Composites

COFs, the metal-free counterparts of MOFs, are freshly created organic polymers that have generated significant enthusiasm among researchers aiming to harness their potential for drug delivery. COFs, created through the covalent bonding of molecules, constitute a new class of porous materials. The crystals of COFs consist only of light atoms, including C, H, O, N, and B [37], therefore mitigating the potential toxicity linked to the metal ions present in their MOF counterparts. They can also be used as drug delivery vehicles. 

Consequently, the number of studies on COFs is increasing [91,92]. Chen et al. created a water-dispersible multifunctional CUF@IR783 composite using cyanine as a stabilizer [93]. It might be contended that the COF@IR783 dispersion developed was suitable for in vivo use. Improved photoacoustic imaging and PTT capabilities in NIR were also present in the dispersion. It can bind precancerous medicines due to its special biodegradability and π–π interactions, which makes COF composites excellent for drug administration. Both in vitro and in vivo tests yielded remarkable results concerning biocompatibility and biodegradability. Still, the application of COFs must be thoroughly tested for toxicological effects, such as long-term systemic toxicity, biosafety, and biodegradation kinetics, in line with FDA rules [94,95].

## 4. Biomedical Applications: Drug Delivery Therapeutics

### 4.1. Drug Delivery

Drug delivery involves encapsulating or loading therapeutic chemicals, particularly intractable and unstable medicines, into nanocarriers for successful transport to the desired target. This method reduces systemic side effects, extends the half-life of unbound pharmaceuticals, and improves the effectiveness of current medications [96,97]. Therefore, it is essential to construct drug delivery carriers with a significant surface area, high drug loading capacity, acceptable biocompatibility, and multifunctional properties. Synergistic PDT has been studied using dual-modal PTT and PDT methods based on nanomaterials. These approaches can affect the immune response or serve as carriers for different therapies and immunotherapeutic drugs to reach a specific target concurrently [98,99]. Because of their NIR light absorption capabilities and photothermal conversion capabilities, graphene, carbon dots, carbon nanotubes (CNTs) [100], and other nanomaterials like copper sulphide NPs (CuS NPs), iron oxide NPs (IONPs), and gold NPs (Au NPs) [35,101,102] have generally been investigated for synergistic PTT and PDT therapies.

MOFs and COFs are used for drug delivery owing to their elevated porosity, ease of synthesis, configurable composition and structure, adjustable size, programmable surface functionality, and biodegradability, as earlier mentioned [103,104]. The porous and structured framework, customizable dimensions, and increased surface area ratios effectively load diverse cargos into MOFs, improving cargo capacity and rendering them suitable for biomedical applications [105,106].

Their highly porous architecture and multifunctionality facilitate the accommodation of substantial quantities of medicinal and imaging chemicals, enabling regulated release while protecting against enzymatic degradation and self-quenching in biological systems [96,97]. Compared with conventional nanomaterials such as inorganic zeolites, mesoporous silica, quantum dots, metal NPs, and organic nanocarriers comprising lipids or polymers, MOFs often exhibit a bigger cargo loading capacity, good biocompatibility, and ease of functionalization [107,108]. Moreover, their gentle synthetic conditions facilitate the development of various MOFs and the integration of a wide array of molecular capabilities on both their internal and external surfaces, encompassing imaging modalities, therapies, and targeting ligands. Many PS-MOF- and PS-COF-based composites have also been employed as targeting agents in the NIR in PTT [109,110]. Table 1 summarizes a few illustrations of PS-COFs and PS-MOFs using different PS groups reported to deliver therapeutic agents.

### 4.2. Principle of Operation for PDT

In PDT using photo-responsive PS-MOF/COF composite (Figure 4), electrons that are not stable in the S_1_ state give off energy as fluorescent quanta. These electrons then move to a more stable excited state (T_1_). When the photosensitive PS-MOF/COF is in the T_1_ state, it creates cytotoxic ROS through two reactions [120]. The PS-MOF/COF reacts directly with the cancerous substance in the type-I reaction, creating free and anion radicals through electron or hydrogen transfer. This creates ROS, which includes hydrogen peroxide (H_2_O_2_), hydroxyl radicals (OH), and superoxide anion radicals (O_2_^−^) [119,121]. In type-II reactions, the PS-MOF/COF directly transfers its energy from the T_1_ state to the fundamental energetic state of O_2_. Subsequently, it produces highly reactive ^1^O_2_ species [31,122]. The characteristics of nanoconjugates formed by MOFs or COFs with PS and their influence on PDT effectiveness have been assessed in several research investigations due to their significance in PDT as optimal organic PS carriers and delivery agents. The MOF nanoparticles can adsorb the PS molecules onto their surface.

#### Photodynamic Therapy

In PDT, non-toxic PS localized in tumors is activated by specific light, transferring energy to adjacent O_2_ molecules and producing ^1^O_2_. This reduces tumor, vascular damage, localized acute inflammation, and immune response. PS-MOF hybrid complexes have lately garnered attention as potential PS for PDT. The hypoxic conditions commonly present in solid tumors diminish the efficacy of PDT. PS, such as Pcs/porphyrin, can convert molecular O_2_ into ^1^O_2_ upon light exposure, enhancing PDT’s therapeutic efficacy [31].

Recent studies have identified nMOFs as a promising element for PDT due to their excellent biocompatibility, biodegradability, suitable dimensions, and capacity for precise functionalization [123]. Consequently, employing PS-MOF hybrid complexes PDT can enhance the effective transformation of ambient tissue oxygen into cytotoxic ^1^O_2_, elevating the cellular temperature [124]. The effective photothermal conversion results in heightened cell necrosis and apoptosis, rendering hybrid composites based on PS-MOFs proficient oxygen sensors within cells [124,125].

A study examined the design and construction of a multifunctional nano platform for catalytic cascades-enhanced PDT, utilizing a combination of p-MOF (UIO) loaded with CaO_2_ nanoparticles, polydopamine (PDA), and platinum precursors aimed at mitigating hypoxia and enhancing the efficacy of PDT in cancer treatment [126]. The research indicated that the UIO@Ca-Pt nano platform could mitigate hypoxia, generate oxygen, facilitate the generation of cytotoxic ^1^O_2_ utilizing PS TCPP under laser irradiation, leading to enhanced anticancer effects in vitro and in vivo and improve the efficacy of PDT in cancer treatment.

The UIO@Ca-Pt nano platform demonstrates outstanding therapeutic efficacy in mice with tumors. Additionally, the findings indicate that the UIO@Ca-Pt NPs exhibited a unique structure, enhanced PDT efficacy, and augmented oxygen-generating capabilities. Moreover, the NPs exhibited much greater anticancer efficacy and reduced hemolytic potential compared to the intermediates and raw materials.

Developing a single-light-source-activated photosensitive material capable of delivering PDT and imaging is highly desirable and worth investigating. To this end, researchers have envisioned synthesizing p-MOFs coated with a thin layer of manganese dioxide nanosheets (MMNPs) utilized as PS for PDT upon NIR activation. Di Zhang and colleagues [127] developed an O_2_-evolving PDT nanoparticle that addresses the limitations of conventional PDT, including inadequate targeting and diminished therapeutic efficiency resulting from tumor hypoxia. The porphyrin-based MOFs were initially synthesized using a modified microemulsion templating method. The cell membrane serves as the shell for MMNPs, enabling cancer cell targeting capabilities (CM-MMNPs). The results demonstrated that the synthesized nanoparticle, CM-MMNPs, displayed excellent colloidal stability, robust targeting efficacy for homologous cancer cells, and dual-mode imaging capabilities (MRI and fluorescence).

Enhanced PDT properties of CM-MMNPs, exhibiting a ^1^O_2_ generation rate of 0.048 min^−1^ were obtained. The results indicate that CM-MMNPs had a pronounced selectivity for homologous cancer cells, and the MnO_2_ layer may be eroded within tumor cells, facilitating fluorescence imaging and disease diagnosis. In vitro, CM-MMNPs also exhibited favorable biocompatibility and antiproliferative properties. The results indicate the potential for employing PS-MOF hybrid composites in a combinatorial strategy for cancer treatment. As stated earlier, a hypoxic tumor microenvironment is a significant barrier to effective PDT. Functional PS-COFs with the potential to improve PDT performance by remodeling the tumor extracellular matrix were created by Zhang et al. [92]. Following intravenous injection, the composites efficiently restored the oxygen-depleted microenvironment of the tumor by accumulating and releasing anti-fibrotic drugs at the tumor site. Furthermore, induced remodeling of the tumor extracellular matrix improved tumor cells’ uptake of proto-porphyrin IX (PPIX)-conjugated peptide NPs (NM-PPIX). Overall, it improved the response to PDT in a synergistic manner.

### 4.3. Principle of Operation for PTT

The operational principles of photothermal PS-MOF/COF-based compounds are intricate and varied. In most photothermal COFs/MOFs containing PS, the S_1_ state typically experiences non-radiative vibrational relaxation, reverting to the ground state through collisions between chromophores and the surrounding biological milieu, thereby dissipating energy as heat [31,116,122,128]. Conversely, high carrier-dose materials, including semiconductors, metals, metal oxides, and quantum dots, can exhibit a photothermal effect via localized plasmon surface resonance [129,130]. When this collective oscillation of electrons diminishes through non-radiative transition, energy is released as heat. In semiconductors with low electron density, thermal energy is produced by recombining electron-hole pairs. Irradiating them will elevate their electrons to a higher energy state in the conduction band, creating a vacancy in the valence band. The electrons and holes will dissipate energy as heat, transitioning to the band edges via vibrational relaxation, recombining near the band edge, and producing further heat through crystal lattice vibrations [131,132].

#### PS-MOF/COF-Based Combination Therapy (PDT, PTT, and Imaging) for Cancer Treatment

Innovative combination treatments are being formulated to enhance cancer treatment. Phototherapy demonstrates distinct advantages in therapeutic applications, including remote controllability, improved selectivity, and reduced biotoxicity compared to chemotherapy. Fluorescence imaging is an effective modality for both in vitro and in vivo applications due to its non-invasive nature and high signal sensitivity [133]. Fluorescence imaging depends on the characteristics of fluorophores that absorb photon energy within a specific wavelength range and subsequently emit photon energy at longer wavelengths. Nanomaterial-based cancer therapy has several advantages over free drugs, including targeted delivery. Targeted delivery targets particular cancer cells precisely, which can be accomplished through active or passive targeting [134].

Numerous advancements have been made in developing high-resolution imaging agents, such as polymeric NPs, lipid-based nanomaterials, metallic and magnetic nanomaterials, and quantum dots, to name a few. These biomedical imaging probes are extensively studied because of their unique optical and electronic properties. The number of approved nanomedicines has not grown substantially despite the abundance of research literature on cancer imaging-related therapies. The primary causes are either biocompatibility issues or a failure to apply the most recent scientific findings. 

For instance, QDs, semiconductor crystallites with a typical size of nanometres, are widely employed to increase the effectiveness of fluorescent markers in biological imaging [135]. The absence of a standardized procedure for producing high-quality QDs and their precise response mechanism and creation process are significant barriers to the clinical translation of QDs [136]. More investigation into targeted drug delivery via nanocarriers is required to enhance clinical translation. To enhance safety and therapeutic efficacy, imaging-guided therapy is crucial as it incorporates visual information to infer the distribution and metabolism of the probe.

PDT, PTT, and imaging are appealing approaches because they are not overly invasive, are simple to administer, and have low systemic toxicity and adverse effects. The processes responsible for the synergy between PDT and PTT can be examined at cellular and tissue levels. Regarding the latter, PTT can target hypoxic tumor locations that may be resistant to the conventional oxygen-dependent PDT. Additionally, PTT can potentially cause additional cell death if the local oxygen levels are low after PDT [137]. The efficacy of bioimaging/PTT/PDT and photostability can be enhanced by integrating PS in MOFs and COFs [138,139].

To make phototherapy more practicable, cancer theranostics must execute simultaneous PDT in a simple, secure, and effective method. Dianwei Wang, Zhe Zhang, and others synthesized p-COF-based NPs (COF-366 NPs) for photoacoustic imaging-guided photodynamic and photothermal cancer therapy under a single wavelength light source [140] (Figure 5). Even with big tumors, COF-366 NPs provided PTT and PDT with a single wavelength light source.

Biodegradability, hemocompatibility, and photostability were excellent for COF-366 NPs. They produced ROS under laser irradiation, causing 4T1 cell death and cytotoxicity. COF-366 NPs aggregated in tumors and suppressed them during laser irradiation. CoF-366 NPs generated ROS in cells and had photothermal activity, enabling PDT and PTT. The combination of PDT and PTT actions of COF-366 NPs completely inhibited big tumors in 4T1 tumor-bearing mice. The study found that COF-366 NPs can completely inhibit big tumors in 4T1 tumor-bearing mice with a single injection and light source. In their cellular tests, the scientists added vitamin C to prevent the PDT effect and decreased the ambient temperature to prevent the PTT impact. Compared to either PDT or PTT alone, it was discovered that the combination of PDT and PTT therapy had a noticeably stronger inhibitory effect on cell viability. With their wide absorption range, which reaches the near-infrared spectrum, the COF-366 NPs in this system may be capable of photoacoustic imaging. This finding provides a novel design concept for porphyrin-based COFs’ biomedical uses, showing that they can completely inhibit tumors when exposed to a single light source.

In a similar study, Hui Zhang, Yu-Hao Li, Yang Chen, and associates [141] developed a novel core–shell nanocomposite for dual-modality imaging-guided PTT and PDT, noted for its minimal cytotoxicity and biotoxicity. During its formation, the Fe_3_O_4_@C@PMOF nanocomposite was synthesized by incorporating a p-MOF into the Fe_3_O_4_@C core. In vivo investigations were conducted on female BALB/c-nu mice bearing MCF-7 breast cancer xenografts. The Fe_3_O_4_@C@PMOF nanocomposite showed considerable tumor accumulation, effective cancer treatment, and low harm to healthy tissue. The resulting nano platform demonstrated dual-modality imaging-guided PTT and PDT treatment capabilities. When exposed to 655 nm laser irradiation, the Fe_3_O_4_@C@PMOF nanocomposites showed high ^1^O_2_ production efficacy, with an estimated quantum yield of 44.38% along with excellent biocompatibility and endurance; nevertheless, the combination of PDT and PDT yielded improved therapeutic efficacy relative to monotherapy.

By integrating multiple treatment modalities into a single structure, the advantages of each are combined while mitigating their drawbacks, opening up new possibilities for the biomedical applications of PS-based MOFs and COFs. The successful application of this system once again highlights the potential of porphyrin-based COFs to treat various diseases effectively through simultaneous PDT and PTT mechanisms. These studies highlight that PS-based COFs and MOFs are not just single PS but also highly effective carriers for drug molecules in addition to being single PS.

## 5. Conclusions and Prospects

This review encapsulates a collection of PS-MOF/COF hybrids and suggests their promising applications in photodynamic therapy, photothermal treatment, and imaging agents. Both COF and MOF exhibit crystalline and porous structures; they can be engineered with tailored features and may undergo post-synthetic modification—advantages not commonly associated with other nanomaterials. In contrast to traditional nano-based materials, COFs and MOFs possess significant potential in biomedicine and demonstrate several advantages, such as excellent biocompatibility, minimal toxicity, and improved photothermal conversion. These hybrid nanocarriers can generate singlet oxygen due to their PS base in photodynamic therapy, eliminating adjacent cancer cells by producing harmful singlet oxygen. Furthermore, incorporating COF/MOF carriers into the PS diminishes self-aggregation and quenching of the PS, significantly improving their efficacy for phototherapy and bioimaging. The porous structure and readily adjustable characteristics of COFs and MOFs render them an exceptional platform for imaging-guided and combination therapies. Nevertheless, hybrid materials comprising MOFs and COFs in biological research remain nascent compared to traditional nanomaterials like mesoporous silica. Numerous difficulties remain to be resolved. The complex COF and MOF materials preparation processes and their extended biocompatibility significantly limit their practical application. Consequently, substantial efforts must be dedicated to designing and fabricating PS-based MOFs and COFs with specific shapes and optimal characteristics. Moreover, the promising performance of MOFs in biological applications is mitigated by the instability of certain MOFs and the toxicity associated with many metal centres. Ultimately, any MOFs or COFs intended for clinical use must comply with regulatory organizations’ rigorous safety and efficacy standards. We assert that the intelligent self-assembly strategy, the incorporation of targeted imaging, and the combined PTT/PDT therapeutic impact offer a progressive methodology for developing more sophisticated platforms to combat bacterial infections.

## Figures and Tables

**Figure 1 molecules-30-00884-f001:**
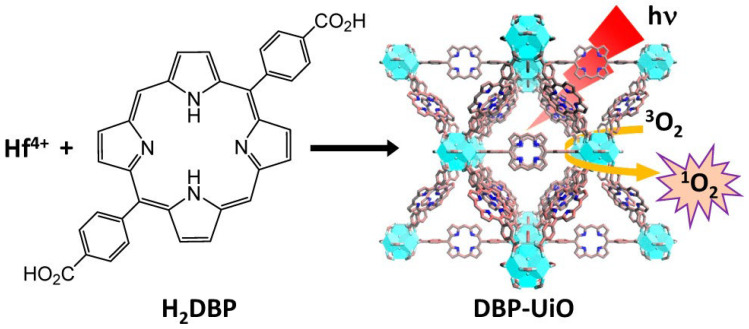
DBP-UiO MOF synthesis, structure, and ROS generation demonstrate that H2DBP molecules are both tethered guests in the MOF channels and MOF organic linkers in the MOF structure. Hf^4+^ metal nodes (Blue) with H2DBP linker in between the metal nodes. *hv* light source (Red). (reproduced under CC BY https://acsopenscience.org/researchers/open-access/, accessed 5 January 2025 [53]).

**Figure 2 molecules-30-00884-f002:**
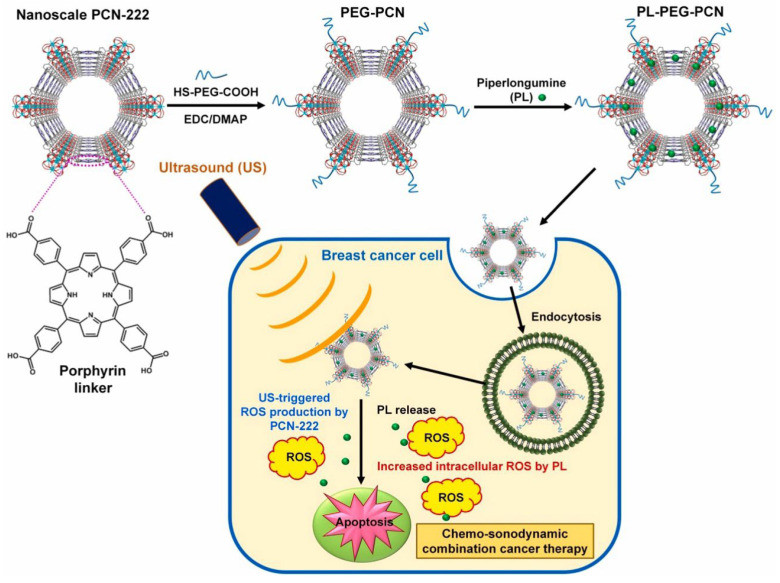
Schematic illustration of the fabrication and SDT effects of pro-oxidant PL-loaded, PEG-coated PCN-222 (PL-PEG-PCN), a nanoscale zirconium-based porphyrinic metal-organic framework (MOF). The diagram’s arrows show the stages involved, including ROS generation by US-irradiated PCN-222, and PL triggers apoptosis in breast cancer cells, enabling chemo/son dynamic combination cancer therapy. Reproduced with permission from [55]. Copyright (2025), Elsevier.

**Figure 3 molecules-30-00884-f003:**
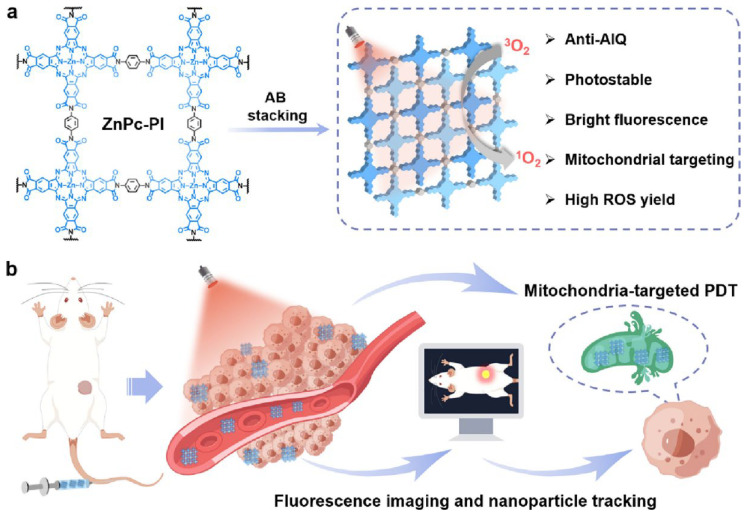
(**a**) Scheme showing the structure and multifunctional performances of ZnPc-PI COF. (**b**) Fluorescence imaging and mitochondria-targeted PDT with ZnPc-PI. Blue arrows show a cross-section of mitochondria targeting during PDT within the cell. Reproduced with permission from [73]. Copyright (2025), American Chemical Society.

**Figure 4 molecules-30-00884-f004:**
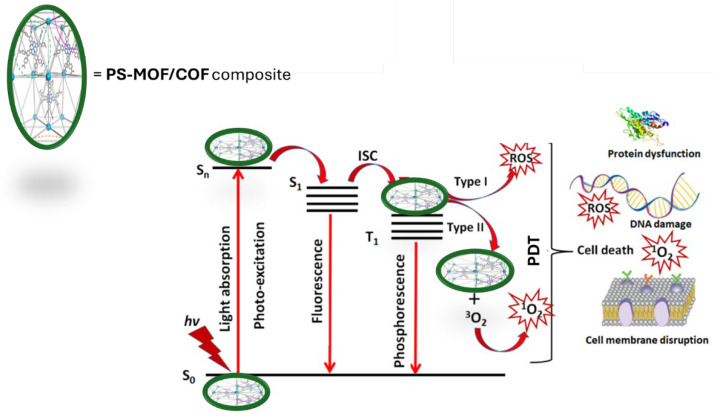
Jablonski diagram demonstrating how a PS-MOF/COF composite can transition from its ground state (S_0_) to a singlet excited state (S_n_) by absorbing light (*hv*) of a certain wavelength and undergo intersystem crossing (ISC) to generate reactive oxygen species (ROS).

**Figure 5 molecules-30-00884-f005:**
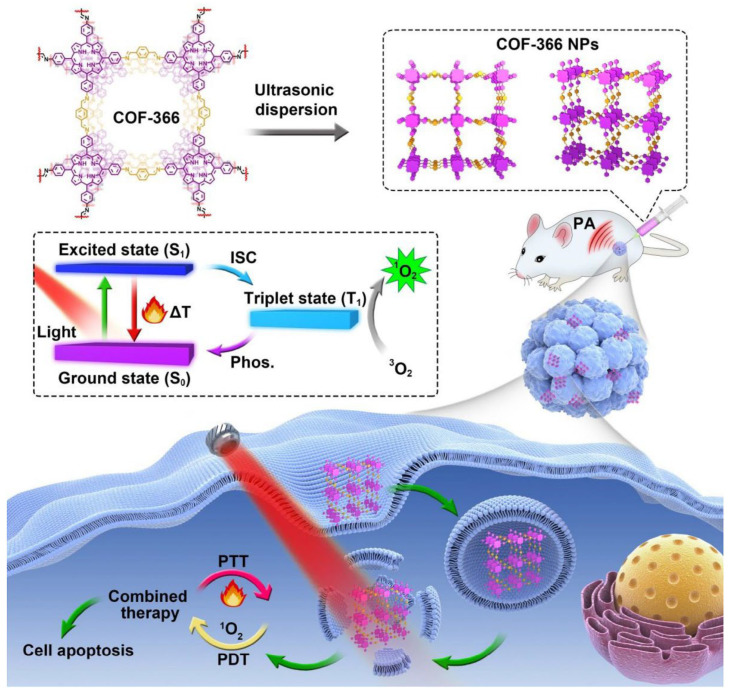
Schematic diagram of combined PDT/PTT therapy for COF-366 NPs. Laser source (Red), COF-366 (Purple), Tumor cell (Blue). Reproduced with permission from [140]. Copyright (2024), Elsevier.

**Table 1 molecules-30-00884-t001:** Typical examples of PS-based MOF/COF composites for cancer therapeutics.

MOFs/COF	PS-Agent	Targeting	Refs
UiO-AM@BODIPY (MOF)	BODIPY	EPR (passive), stimuli (pH-responsive)	[111]
ACF@PCN-222@ MnO_2_-PEG (APM), (MOF)	Porphyrin	Stimuli (hypoxic, H_2_O_2_- triggered drug release)	[112]
ZnP@Hf-QC (MOF)	Pc	^1^O_2_ responsive upon 700 nm light irradiation	[109]
LZU-1-BODIPY-2I/ZU-1-BODIPY-2 (COF)	BODIPY	^1^O_2_ generation	[113]
UCCOFs-1	Porphyrin	NIR luminescence imaging and ^1^O_2_ generation	[114]
CG@COF-1@PDA	ICG	^1^O_2_ and hyperthermia-generating abilities	[115]
VON@COF-Por	Porphyrin	^1^O_2,_ 808 nm (PTT), MCF-7 tumor	[116]
PCN-222-SO_3_H (PCN-SU) (MOF)	Porphyrin	EPR (passive)	[117]
UiO-PDT (MOF)	BODIPY	^1^O_2_ generation	[118]
COF-618-Cu	Porphyrin	^1^O_2_+ PTT, immunogenic cell death	[119]

## Data Availability

Not applicable.

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
