# Peer review of "Hybrid Nanoplatforms Based on Photosensitizers and Metal/Covalent Organic Frameworks for Improved Cancer Synergistic Treatment Nano-Delivery Systems"

_molecules, 2025, doi:10.3390/molecules30040884_

Round 1

Reviewer 1 Report

Comments and Suggestions for Authors

This review provides an overview of recent progress in photosensitizers integrated within MOF or COF, covering synthetic methods, biocompatibility, and biomedical applications. This topic is interesting, however, the above-mentioned issues should be considered before publication.

1.      The title “Hybrid Nanoplatforms Based on Photosensitizers and Metal/Covalent Organic Frameworks for Improved Photodynamic Therapy Nano-Delivery Systems” is refined within Photodynamic Therapy, bioapplication like photothermal therapy or imaging is not covered. In addition, the picture listed should highlight this topic. Since Figure 1 is chemoradiation, Figure 2 listed PS s undergoing clinical studies, these molecular structures are not adapted to construct  PS-MOF/COF composites.

2.      As stated by the author, “Metal-organic frameworks (MOFs) and covalent organic frameworks (COFs) represent two relatively new material types. These materials have high surface areas and permanent porosity~”. These two materials share similar properties and differences, and discussion associates should be strengthened.

3.      The Figure legend should be formatted, especially for Figure 1 and Figure 4. Besides, pay attention to the cited references to ensure their consistency.

4.      In section 3 “Biocompatibility”, the subtitle “MOFs”, “COFs”, and “PS” are separated. Especially, discussions about the PS-MOF/COF composites are missing.

5.      The discussion is superficial, it is suggested to supply a comprehensive comparison of PS-MOF/COF nanocomposites and provide an overview of strategies for improved PDT.

Comments on the Quality of English Language

The English could be improved to more clearly express the research.

Author Response

This review provides an overview of recent progress in photosensitizers integrated within MOF or COF, covering synthetic methods, biocompatibility, and biomedical applications. This topic is interesting. However, the above-mentioned issues should be considered before publication.

  1. The title “Hybrid Nanoplatforms Based on Photosensitizers and Metal/Covalent Organic Frameworks for Improved Photodynamic Therapy Nano-Delivery Systems” is refined within Photodynamic Therapy, bio-application like photothermal therapy or imaging is not covered.

Repl: Thank you, the title is now re-adjusted to accommodate all the discussed bio-applications.

 In addition, the picture listed should highlight this topic. Since Figure 1 is chemoradiation, Figure 2 listed PS s undergoing clinical studies, these molecular structures are not adapted to construct  PS-MOF/COF composites.

Reply: Thank you for your correction; adjustments have been made to correct the mistake

  1. As stated by the author, “Metal-organic frameworks (MOFs) and covalent organic frameworks (COFs) represent two relatively new material types. These materials have high surface areas and permanent porosity~”. These two materials share similar properties and differences, and discussion associates should be strengthened.

Reply: Thank you. Additional information regarding this has now been added.

  1. The Figure legend should be formatted, especially for Figure 1 and Figure 4. Besides, pay attention to the cited references to ensure their consistency.
  2. In section 3 “Biocompatibility”, the subtitle “MOFs”, “COFs”, and “PS” are separated. Especially, discussions about the PS-MOF/COF composites are missing.

Reply: The mistake has been corrected, thank you.

  1. The discussion is superficial, so it is suggested that a comprehensive comparison of PS-MOF/COF nanocomposites be supplied, as well as an overview of strategies for improved PDT.

Reply: Additional insights are now included and discussed; thank you.

Reviewer 2 Report

Comments and Suggestions for Authors

In this reveiw, the authors report the preparation of nanohybrids with photosensitizers adsorbed on the surface or embedded in MOFs/COFs and of MOFs with photosensitizers as linkers. The use of these nanoparticles for cancer treatment (PDT, PTT) or for bio-imaging is described. From a general point of view, the topic is of interest for the readers of Molecules. However, in the present state, I can not recommend this manuscript for publication. Here are my comments :

- the authors chose to consider only a relatively limited number of articles on the subject. On what basis was this choice made?

- the advantages and drawbacks of different nanohybrids developed for cancer or imaging applications are not discussed. This should markedly be improved.

- the optical properties and anti-cancer efficacy of nanohybrids should also be compared with molecular photosensitizers. This would highlight the advantages of these nanohybrids.

- line 143 + line 333 : clarify the legend. Do not simply provide the reference.

- line 299 : correct hydroxyl radicals (.OH) and superoxide radicals (O2.-).

- the language could be improved. Some parts of the manuscript contain many repetitions.

Comments on the Quality of English Language

see my comments above. The text must be clarified.

Author Response

- the authors considered only a relatively limited number of articles on the subject. On what basis was this choice made?,

Reply: the literature was limited to recently published articles presenting at least different PS derivatives for COF and MOF synthesis.

- the advantages and drawbacks of nanohybrids developed for cancer or imaging applications are not discussed. This should markedly be improved.

Reply: A section on this is now included, thank you.

- the optical properties and anti-cancer efficacy of nanohybrids should also be compared with molecular photosensitizers. This would highlight the advantages of these nanohybrids.

Reply: This is now discussed in the manuscript; thank you.

- line 143 + line 333: clarify the legend. Do not simply provide the reference.

Reply: the mistake is now corrected, thank you.

- line 299: correct hydroxyl radicals (.OH) and superoxide radicals (O2.-).

Reply: the error is now corrected, thank you.

- the language could be improved. Some parts of the manuscript contain many repetitions.

Reply: The manuscript has been edited. Thank you.

Round 2

Reviewer 1 Report

Comments and Suggestions for Authors

Most of the list questions have not been fully addressed. The English should be improved to more clearly express this research.

Comments on the Quality of English Language

The English should be improved to more clearly express this research.

Author Response

We are so sorry for the mistake; an error was made during submission.

Reviewer 2 Report

Comments and Suggestions for Authors

Most of my comments were considered.

The manuscript can be accepted by Molecules.

Comments on the Quality of English Language

The language could be improved.

Author Response

In this reveiw, the authors report the preparation of nanohybrids with photosensitizers adsorbed on the surface or embedded in MOFs/COFs and of MOFs with photosensitizers as linkers. The use of these nanoparticles for cancer treatment (PDT, PTT) or for bio-imaging is described. From a general point of view, the topic is of interest for the readers of Molecules. However, in the present state, I can not recommend this manuscript for publication. Here are my comments :

  • the authors considered only a relatively limited number of articles on the subject. On what basis was this choice made?

Thank you for your correction. We have added more articles on the subject.

  • The advantages and drawbacks of nanohybrids developed for cancer or imaging applications are not discussed. This should be markedly improved.

Thank you. We have now added some information regarding other nanohybrids without changing the course of the manuscript.

Reply: A section on this is now included, thank you.

- the optical properties and anti-cancer efficacy of nanohybrids should also be compared with molecular photosensitizers. This would highlight the advantages of these nanohybrids.

- line 143 + line 333: clarify the legend. Do not simply provide the reference.

Reply: the mistake is now corrected, thank you.

- line 299: correct hydroxyl radicals (.OH) and superoxide radicals (O2.-).

Reply: the error is now corrected, thank you.

- the language could be improved. Some parts of the manuscript contain many repetitions.

Reply: The manuscript has been edited. Thank you.